# Failure of Diphtheria Toxin Model to Induce Parkinson-Like Behavior in Mice

**DOI:** 10.3390/ijms22179496

**Published:** 2021-08-31

**Authors:** Lucie Valek, Irmgard Tegeder

**Affiliations:** Institute of Clinical Pharmacology, Medical Faculty, Goethe-University, 60590 Frankfurt, Germany; valek@med.uni-frankfurt.de

**Keywords:** Cre-recombinase, dopamine transporter, SLC6a3, diphtheria toxin, tyrosine hydroxylase, open field behavior, motor functions

## Abstract

Rodent models of Parkinson’s disease are based on transgenic expression of mutant synuclein, deletion of PD genes, injections of MPTP or rotenone, or seeding of synuclein fibrils. The models show histopathologic features of PD such as Lewi bodies but mostly only subtle in vivo manifestations or systemic toxicity. The models only partly mimic a predominant loss of dopaminergic neurons in the substantia nigra. We therefore generated mice that express the transgenic diphtheria toxin receptor (DTR) specifically in DA neurons by crossing DAT-Cre mice with Rosa26 loxP-STOP-loxP DTR mice. After defining a well-tolerated DTx dose, DAT-DTR and DTR-flfl controls were subjected to non-toxic DTx treatment (5 × 100 pg/g) and subsequent histology and behavioral tests. DAT protein levels were reduced in the midbrain, and tyrosine hydroxylase-positive neurons were reduced in the substantia nigra, whereas the pan-neuronal marker NeuN was not affected. Despite the promising histologic results, there was no difference in motor function tests or open field behavior. These are tests in which double mutant Pink1^−/−^SNCA^A53T^ Parkinson mice show behavioral abnormalities. Higher doses of DTx were toxic in both groups. The data suggest that DTx treatment in mice with Cre/loxP-driven DAT-DTR expression leads to partial ablation of DA-neurons but without PD-reminiscent behavioral correlates.

## 1. Introduction

A number of rodent models of Parkinson’s disease based on human genetics of PD have been described [1], and have been extensively studied in terms of behavior, mostly motor functions and PD-like histopathology [2,3]. Behavioral studies of non-motor manifestations such as cognition, olfaction, anxiety-like behavior and gastrointestinal functions are fragmented. The majority of genetic PD rodents carry human mutant alpha synuclein (SNCA) [4,5], which causes early onset dominant PD in humans. Other models are knockouts of genes associated with early onset recessive PD such as Pink1 [6,7,8] and Parkin [9]. Transgenic mutant LRRK2 models have been introduced to study late onset dominant or sporadic PD [10]. While histology revealed synuclein aggregates in aged to old mice in genetic PD models, motor manifestations were mostly subtle [6,7,8,9,10].

In addition to genetic PD models, neurotoxin-based models evoked by 1-methyl-4-phenyl-1,2,3,6-tetrahydropyridine (MPTP), 6-hydroxydopamine (6-OHDA), paraquat and rotenone [11] are used either to evoke a local lesion of the substantia nigra via stereotaxic injection (MPTP or 6-OHDA) or to cause systemic toxicity by repeated intraperitoneal injections [2,3]. The toxic metabolite of MPTP, MPP+, is taken up by DA neurons via the dopamine transporter (Slc6a3/DAT) [12], conferring preferential toxicity to DA neurons [13]. Owing to the limitations of the hazardous MPTP, the chronic rotenone model was developed [11].

MPP+ and rotenone block complex-I of the mitochondrial respiratory chain and address the mitochondrial aspect of PD pathophysiology. These models do not address the aspect of protein aggregates and prion-like spreading of pathological SNCA [14]. Therefore, mutant SNCA fibrils were seeded into the olfactory bulb or muscle to observe the spreading of SNCA aggregates [15,16]. SNCA spreading is also used to assess the prion-like spreading of glial SNCA inclusions to model the pathology of multiple system atrophy (MSA) [17], called MSA-prions [18]. MSA is a rapidly progressive synucleopathy arising from the misfolding and accumulation of SNCA species, mainly in oligodendrocytes [19], but transgenic models directing mutant SNCA to oligodendrocytes replicate only some aspects of human MSA [20,21], likely owing to complex genetics [22,23] and confounding lipid mediators such as glucosylceramides [24,25,26,27,28] and monounsaturated fatty acids [29,30,31] that can amplify the pathology of SNCA [32].

Single models have limitations, but all have contributed to the increasing knowledge of PD pathophysiology in recent years. There is still a caveat of models that are non-invasive and lead to a slowly progressive predominant loss of DA neurons not locally restricted to a single toxin injection site and not associated with substantial systemic toxicity, and that do not rely on the transgenic manipulation of a single gene.

Diphtheria toxin-based models have gained some popularity to ablate specific subsets of cells where the transgenic expression of the simian diphtheria toxin receptor (DTR; gene HBEGF: heparin binding epidermal growth factor-like) is driven by the promoter of a cell type-specific gene. Alternatively, the transgenic DTR is activated in specific cells by using the Cre/loxP method. For the latter, the respective Cre-mouse is crossed with a mouse carrying transgenic DTR headed with a loxP-STOP-loxP site (LSL-site) and inserted into the Rosa26 locus (R26-LSL-DTR mice). Cre-recombinase-mediated excision of the STOP codon leads to DTR expression in Cre positive cells. Because mouse HBEGF normally does not bind diphtheria toxin, treatment with DTx can be used to specifically ablate the cells that express the transgenic DTR [33]. The Cre/loxP DTR method was successfully used to ablate, for example, T- or B-cells [33,34], dendritic cells or macrophages [35,36], fibroblasts [37] and other peripheral cells. However, DTx badly crosses the intact blood-brain barrier and DTx has been shown to kill DTR-positive mice at doses as low as 100 pg/g [38]. Nevertheless, the techniques have been described to specifically ablate subsets of neurons in the CNS, for example, agouti-related protein (AgRP)-positive hypothalamic neurons [39,40] or ETS domain factor (Pet1)-positive serotoninergic neurons [41]. CXCR3-DTR or MOG-DTR were used to delete microglia or oligodendroglia, respectively [33,42]. The DTx doses and schedules range from 1.4 pg/g up to 250 ng/g per day for 2–10 consecutive days [41,43] (Table 1).

Based on the promising results showing DTx-mediated ablation of neuron subtypes, we set out to test the Cre/loxP-based technique to achieve a non-invasive slowly progressive ablation of DA neurons as a novel PD model. The theoretical advantages include high flexibility of the Cre/loxP technique and dose-dependent adjustments of time courses and severity. We used the dopamine transporter (Slc6a3/DAT) Cre+ mouse to drive DTR expression in DA-positive neurons. The efficacy of DTx was assessed by RNA and protein analysis of DTA and histology of tyrosine hydroxylase in the SN, and by studying behavior in open field and motor tests in comparison with Pink1^−/−^SNCA^A53T^ PD mice [52].

## 2. Results

### 2.1. Partial Ablation of Tyrosine Hydroxylase-Positive Neurons after Low Dose DTx

We generated DAT-DTR mice by crossing DAT-Cre (Slc6a3-Cre) with R26-LSL-DTR mice. Successful recombination was confirmed by genotyping of the DTR transgene and Cre-recombinase (Figure 1A). To define a safe, well-tolerated dosing schedule, single doses were tested in 1–2 mice (Appendix A) to cover the dosage range used in previous publications (Table 1). The previously used dose of 100 ng/mouse (≈4 ng/g per day) for 1–5 consecutive days was highly toxic (Figure 1B, Appendix A). We therefore used a low dose of 100 pg/g per day for 5 consecutive days (cumulative dose 0.5 ng/g, ≈12.5 ng/mouse) (Figure 1C). At this dose, survival was 100% and body weights were stable.

One month after DTx treatment, DAT mRNA in the midbrain was highly variable (Figure 1D), but DAT protein was reduced (Figure 1E,F).

Tyrosine hydroxylase immunoreactive neurons in the SN were assessed by immunostaining in mice who received 5 × 100 pg/g DTx (Figure 2). Quantification of TH revealed a significant loss of TH immunoreactivity in DAT-DTR mice as compared with DTR-flfl control mice (Figure 3A–C), whereas the overall number of neurons as assessed by NeuN staining of neuronal nuclei was not affected (Figure 3A,B,D).

### 2.2. No Difference of Behavior in DTx-Treated DAT-DTR versus DTR-flfl Mice

We assessed behavior in the open field test (OFT, Figure 4A–C) and in motor function tests (Figure 4D,E), which have been successfully used previously to reveal behavioral manifestations of PD in mice [4,52,53,54,55,56]. The OFT was done before and 30 days after DTx treatment. The exploration of the arena was lower in the retest trial in all mice including control mice that received saline injections instead of DTx. As a result, the travel path was shortened in the second test and mice spent less time in the center area, indicating a loss of curiosity about the open arena upon the retesting. However, there was no difference between groups, and the variability was high, particularly at baseline (first test). There were also no differences between groups in the pole test (Figure 4D), the accelerating Rotarod (Figure 4E) or body weights (Figure 4F).

The test-retest changes in the OFT were further assessed as paired data to reveal the individual’s first and second tests and putative learning differences (Figure 5). The “boring“ effect of retesting was obvious in the majority of mice, but without difference between groups.

For comparison of the behavioral data, we assessed behavior in Pink1^−/−^SNCA^A53T^ mice (Figure 6), which represent a well-studied genetic PD model [1,52,57]. These mice showed overactivity in the OFT (longer path) and spent relatively more time in the border zone than the wild-type control mice, showing that they were running along the walls. The comparison reveals that the OFT is a sensitive test for assessment of PD-associated behavioral manifestations in mice. It is one of the most frequently used tests in genetic and toxic PD models [1].

## 3. Discussion

We show in the present study that DTx injections in mice were very toxic at doses used in previous studies targeting peripheral cells, glia or neurons of the central and peripheral nervous system. The mortality was higher in DAT-DTR (Cre+) mice than in DTR-flfl controls, but toxicity was high irrespective of the genotype. The high mortality after injection of single doses agrees with a previous study showing high mortality at doses as low as 0.1 ng/g [38]. In the study of Cha et al., five out of five DTR-transgenic mice died at 0.1 ng/g [38]. In our experiments, this low dose (0.1 ng/g/d for five consecutive days) was safe both in all floxed control mice and in DAT-DTR mice. However, this treatment caused only a moderate loss of TH-positive neurons in the SN, which was statistically significant by quantitative analyses of TH immunoreactivity but did not manifest in behavior reminiscent of PD or resembling previously described PD mice [52].

We searched for putative explanations for the high rate of deaths at higher DTx doses, which have been successfully used in a number of previous studies to ablate neurons or glia in the CNS [33,39,41,42,46,47,50,51,58,59]. DTx is an exotoxin purified from *Corynebacterium diphtheriae* and is provided as lyophilized powder and was reconstituted in sterile water and further diluted in saline. No chemicals were involved. As expected, mouse RAW264.7 macrophages were not stimulated or growth-affected with the stock solution of 0.1 mg/mL (final concentration 0.01 mg/mL) and with working solutions used for injections in mice. The mRNA of inducible nitric oxide synthase remained at baseline, whereas it increased 100-fold upon LPS/IFNγ stimulation, which was used as a positive control. The experiment strongly argues against a relevant bacterial contamination. In agreement, an endotoxin ELISA was negative in working solutions (Appendix A, Appendix A).

DTx mortality did not significantly differ between DAT-DTR mice as compared to DTR-flfl, suggesting that DTx-evoked death was not caused by fast specific cell ablation. We were able to define a dose that was well tolerated (5 × 0.1 ng/g) in all mice and, promisingly, we observed an ablation of TH-positive neurons in the SN that was significant at statistical immunofluorescent analyses. However, the effect was not sufficient to alter mouse behavior in tests that do show PD-associated behavior in genetic Pink1^−/−^SNCA^A53T^ double-mutant PD mice [52]. The result shows that a moderate ablation of DA neurons is insufficient to result in measurable behavioral effects or that the DA neurons in the SN are not the primary driving cause for abnormal behavior in genetic PD mice. This is in agreement with the histologic studies in these mice showing SNCA aggregates throughout the brain and spinal cord not predominantly localized in the SN [57]. Hence, a specific moderate loss of DA neurons does not cause a PD-like disease in mice.

## 4. Materials and Methods

### 4.1. Mouse Model

Mice expressing the simian diphtheria toxin receptor (DTR) specifically in dopaminergic neurons were generated via Cre-loxP-mediated recombination by mating B6-DTR mice (Jackson Stock No: 007900) with DAT-IRES-Cre mice (Jackson Stock No: 006660). B6-DTR mice have the simian diphtheria toxin receptor (simian HBEGF cDNA base pair 56-682; HBEGF = heparin-binding EGF-like growth factor, chimpanzee) inserted into the ROSA26 locus. Specific expression of DTR is ensured by an upstream loxP-flanked STOP sequence (R26-LSL-DTR). When bred to Cre recombinase-expressing DAT-IRES-Cre mice, the STOP sequence is deleted in dopaminergic neurons, leading to DTR expression in DA-neurons, which are susceptible to ablation upon treatment with diphtheria toxin (DTx). Control mice are not sensitive to DTx because the epithelial growth factor receptor in mice does not bind DTx as it does in other species [60].

Mice had free access to food and water and were maintained in climate-controlled rooms with a 12 h light-dark cycle. Behavioral experiments were performed between 10 am and 3 pm. The experiments were approved by the local Ethics Committee for animal research, Darmstadt, Germany (approval #V54—19c 20/15—FK1096, approval date 11 October 2017), adhered to the guidelines for pain research in conscious animals of the International Association for the Study of Pain (IASP) and those of the Society of Laboratory Animals (GV-SOLAS), and were in line with the European and German regulations for animal research.

For DA-neuron ablation, DAT-DTR mice and DTR-flfl controls were injected intraperitoneally (i.p.) with diphtheria toxin (DTx) in saline for five consecutive days. DTx is an exotoxin of *Corynebacterium diphtheriae* (DTx; Sigma, Germany, D0564). The lyophilized powder was reconstituted in sterile water to achieve a stock solution of 0.1 mg/mL DTx. The stock was diluted 1:100–1:2000 in 0.9% saline to obtain working solutions for injections in mice. Control mice received 0.9% saline. For dose-finding studies, mice received low doses of 5 × 100 pg/g/d, or they were treated with 1–5 doses of 50 and 100 ng per mouse (≈2 or 4 ng/g) or single doses of 1.5, 3.1, 6.25, 12.5, 25 or 50 ng/g (Appendix A). The objective was to define a protocol that evoked specific effects in DAT-DTR mice without unspecific toxicity in DTR-flfl control mice. Mice were 8–16 weeks old at the start of the injections and were observed for 1–2 months after the last DTX injection.

### 4.2. Immunofluorescence Analysis

Mice were terminally anesthetized with isoflurane and cardially perfused with cold 0.9% NaCl followed by 2% paraformaldehyde for fixation. Tissues were excised, postfixed in 2% PFA for 2 h, cryoprotected overnight in 20% sucrose at 4 °C, embedded in small tissue molds in cryo-medium and cut on a cryotome at 12 μm. Slides were air-dried and stored at −80 °C. After thawing, slides were immersed and permeabilized in 1x PBS with 0.1% Triton-X-100 (PBST), then blocked with 3% BSA/PBST, and subsequently incubated overnight with the first primary antibody in 1% BSA/PBST at 4 °C. After washing three times with PBS, slides were incubated with the secondary antibody for 2 h at room temperature, followed by 10 min incubation with DAPI and embedding in Aqua-Poly/Mount. Primary antibodies were directed against tyrosine hydroxylase (rabbit, 1:200, Thermo Fisher Scientific, #OPA1-04050) and NeuN (mouse, 1:100, Chemicon, #MAB377). Secondary antibodies were labeled with Alexa488 or Cy3 (Invitrogen, Sigma, Life Technologies, all Germany). Nuclei were counter-stained with DAPI. Slides were analyzed on an inverted fluorescence microscope (BZ-9000, KEYENCE, Neu-Isenburg, Germany).

FIJI ImageJ was used for quantitative assessment of tyrosine hydroxylase-positive DA neurons in the substantia nigra and of all NeuN-positive neurons. Images were converted to 8-bit images, background subtracted, and converted to binary images via threshold setting using the Intermodes algorithm implemented in ImageJ. The area of immunoreactive particles was assessed by using the Particle Counter of ImageJ.

### 4.3. Western Blot

Whole cell protein extracts were prepared in RIPA lysis buffer (Sigma) containing a protease inhibitor cocktail (Roche) and PMSF 10 µg/mL, separated on a 10% SDS-PAGE gel (30 µg/lane), then transferred to nitrocellulose membranes (Amersham Pharmacia Biotech, Freiburg, Germany) by electro-blotting. Blots were blocked in 1:1 Odyssey buffer in PBS and developed in Odyssey buffer in 1x PBS/Tween 20. For detection of specific proteins, blots were incubated with the primary antibody against DAT/Slc6a3. β-Actin was used as a loading control. Secondary antibodies were conjugated with IRDye 680 or 800 (1:10,000; LI-COR Biosciences) and blots were analyzed on the Odyssey Infrared Imaging System (LI-COR Biosciences). The ratio of the respective protein band to the loading control was used for semi-quantitative analysis (Image Studio Light^®^, Odyssey).

### 4.4. Quantitative Real-Time PCR

Total RNA was extracted from stimulated RAW cells with the RNeasy Mini Kit (QIAGEN, Hilden, Germany) and reverse transcribed to cDNA fragments using the Verso cDNA Synthesis Kit (Thermo Fisher Scientific, Germany). The mRNA was amplified and quantified using the SybrGreen detection system on a QuantStudio 5 Real-time PCR System (Thermo Fisher Scientific) using the SybrGreen detection system (Thermo Fisher Scientific). Relative mRNA expression was calculated according to the ΔCt method. Relative values were normalized to the housekeeping gene eukaryotic translation elongation factor-2 (EEF2; forward primer 5’- agg cct gtg taa tat agc tgc g -3′, reverse primer 5′-ctc tgt gta gtt tgt agc tct gtc t-3′). Inducible NOS (NOS2) gene expression was detected with the RT2 qPCR Primer Assay for Mouse Nos2, NM_001313921 (QIAGEN).

### 4.5. Cell Culture and Endotoxin Screening

RAW 264.7 mouse macrophages were grown in VLE-DMEM medium supplemented with heat-inactivated 10% fetal bovine serum at 37 °C and 5% CO_2_ atmosphere in a humidified incubator. To assess unspecific DTx toxicity in mouse cells, RAW cells (150,000 cells/5 cm plate) were stimulated with 100 µL/mL of normal saline, the DTx stock solution (0.1 mg/mL; final concentration 0.01 mg/mL) or DTx working solutions for injections in mice (100 pg/g, 1.5 ng/g). Stimulation of RAW cells with 20 ng/mL lipopolysaccharide (LPS) and 20 ng/mL interferon gamma (INFγ) was used as positive control. After 24 h, cells were washed twice with PBS and harvested for qRT-PCR.

Endotoxin levels in DTx stock and working solutions were determined with the ToxinSensorTM Chromogenic LAL Endotoxin Assay Kit (GenScript, Germany) according to the manufacturer’s protocol. Endotoxins catalyze the activation of a proenzyme in limulus amebocyte lysate (LAL), which cleaves the colorless substrate to generate a yellow end product, which was measured spectrophotometrically at 545 nm. The quantification is based on a standard curve.

### 4.6. Behavioral Analyses

Motor coordination and running performance were assessed with the accelerating Rotarod (16–32 rpm, ramp over 5 min, cutoff 5 min; Ugo Basile, Gemonio, Italy) before and after DTx administration (6–8 mice per group). Mice were trained 2–3× before start. The fall-off time was averaged from 2 tests.

In the pole test, mice were placed on top of a vertical pole with all paws grasping the pole and the head pointing upwards. The pole has a rough surface preventing mice from sliding down. The time required to turn 180° and to reach the floor was recorded. Each mouse performed three consecutive trials with a break of 30 s between the trials.

In the open field test (OFT), mice were placed in the middle of an open field (50 × 50 cm width, 38 cm height) and allowed to move freely for 10 min. Mice were observed with a video camera. Virtual zones were defined as center and border. Locomotion (total paths), visits and times spent in zones were analyzed with VideoMot2, which uses a 2-point tracking (TSE Systems).

### 4.7. Data Analysis and Statistics

Graphpad Prism 8.4 or 9.0 were used for statistical evaluation. Data are presented as box or box/scatter plots. Sample sizes are given in the respective figure legend. Data were analyzed with Student’s *t*-tests (2 groups) or univariate or 2-way analysis of variance (ANOVA) and subsequent post hoc *t*-tests using a Dunnett, or group-vise adjustment of alpha according to Šidák.

## 5. Conclusions

We infer that DTx may cause substantial toxicity in mice and that the Cre/loxP-driven DTR/DTx model is not useful for the in vivo study of PD in mice. Maybe extending the daily injections or repeated cycles of DTx would lead to clearer death of dopaminergic neurons, for example, three cycles and observation for 3–6 months. The Cre/loxP DTR model might still be useful to model PD by using a less specific Cre-mouse such as TH-Cre that would theoretically address the involvement of the autonomic nervous system in PD and other synucleopathies. It is of note that attempts to ablate specific cholinergic neurons with a DTx model for Alzheimer’s disease were not successful to achieve an Alzheimer phenotype [61], although an NGF-DTx conjugate was directly injected into the forebrain [61]. Intracerebroventricular or intra-striatal injection of DTx could also be used in DAT-DTR mice described in our study, but one advantage of the non-invasive model would be lost.

## Figures and Tables

**Figure 1 ijms-22-09496-f001:**
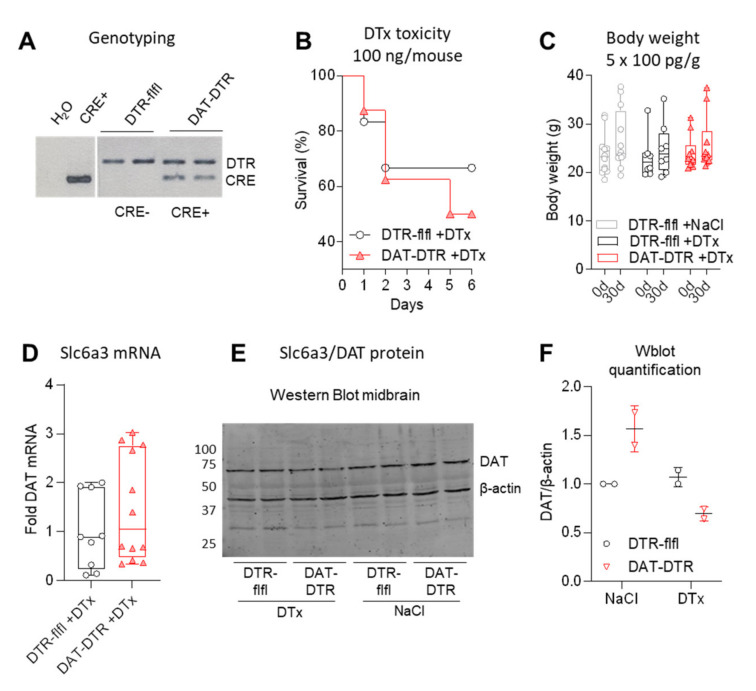
Efficacy of DTx treatment of DAT-DTR mice versus control mice. (**A**) Exemplary agarose gel showing a typical genotyping result. (**B**) Survival curve of mice injected with 100 ng/mouse per dose (≈4 ng/g) DTx. Details of mice and sample sized per dose in Appendix A. (**C**) Body weight at baseline and 30 d after treatment with low non-toxic dose of 100 pg/g/d DTx for five consecutive days (cumulative dose 0.5 ng/g). All mice survived. Each scatter is a mouse (*n* = 12, 8 and 10). (**D**) Quantitative RT-PCR of DAT/Slc6a3 RNA in the midbrain after 5 × 0.1 ng/g DTx treatment (triplicate samples of *n* = 3–4 mice). (**E**,**F**) Western blot analysis and quantification of DAT/Slc6a3 protein in the midbrain after 5 × 0.1 ng/g DTx (example shows *n* = 2 per group). For C–F, mice were 11–13 weeks at the onset of DTX, and tissue was obtained at 30 d.

**Figure 2 ijms-22-09496-f002:**
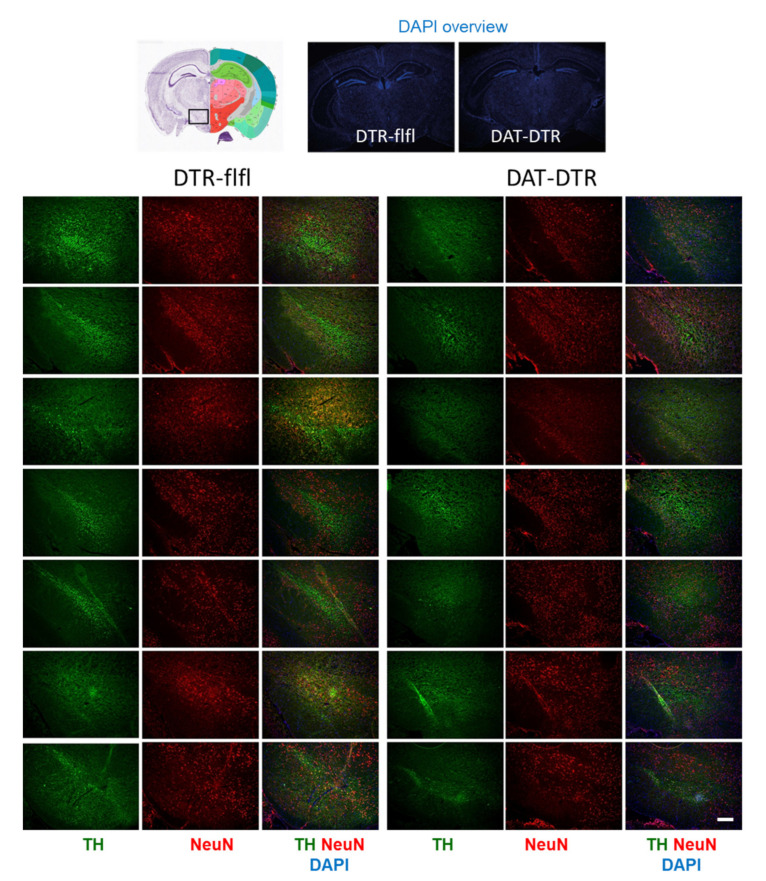
Immunofluorescence analysis of
tyrosine hydroxylase (TH) as a marker for DA neurons in the substantia nigra
(SN). The pan-neuronal marker NeuN was used to label all neurons. DAPI was used
as counterstain of nuclei. Mice were treated with 5 × 0.1 ng/g DTx. Mice were
9–13 weeks old at the onset of DTx and were sacrificed 6–7 weeks after the last
DTx dose. The image shows examples of *n* = 4–6 mice per group (one side
or both sides). Scale bar 200 µm.

**Figure 3 ijms-22-09496-f003:**
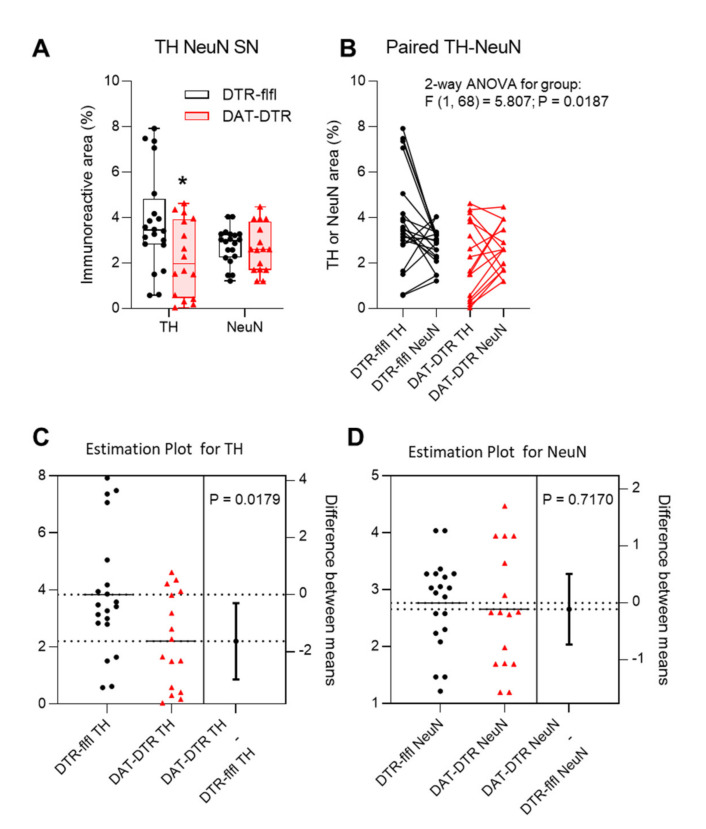
Quantification and statistical comparison of TH and NeuN immunofluorescence. (**A**,**B**) Box/scatter plot and Paired data analysis of the TH and NeuN immunopositive areas determined with the Particle Counter in FIJI ImageJ. The scatter represents images of the left and right SN of 4–6 mice per group. Data of TH and NeuN were compared by ANOVA and subsequent unpaired, two-tailed t-test for TH and NeuN separately. The asterisk shows a significant result with *p* < 0.05. (**C**,**D)** Estimation plots for TH and NeuN showing the group difference obtained via t-tests.

**Figure 4 ijms-22-09496-f004:**
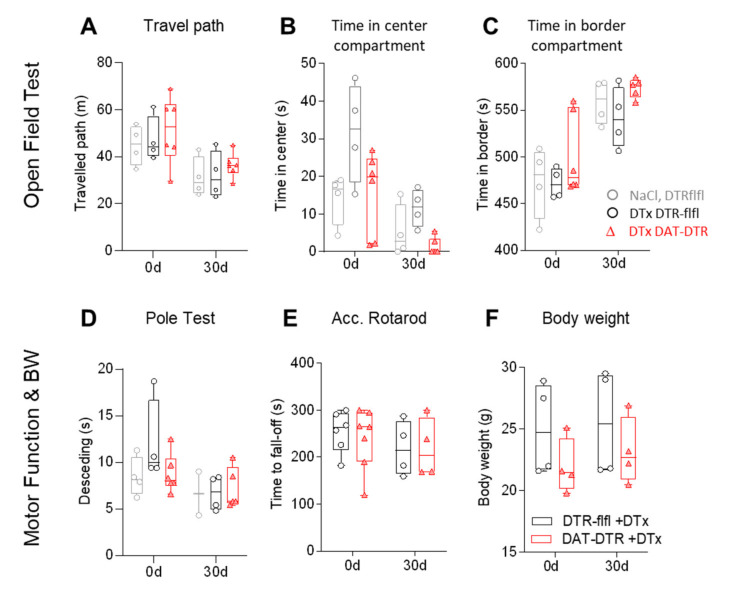
Behavior in DTx-treated mice. (**A**–**C**) Travel paths and the times spent in the center compartment and the border compartment in a classical open field test (OFT). The behavior was tested at baseline (0d) and 30d after DTx treatment with 0.1 ng/g/d for 5 consecutive days. Exploration of the center dropped in all mice in the 30 d retest without difference between groups. Each scatter is a mouse, sample size *n* = 4–6. (**D**–**F**) Motor coordination analysis in the pole test and accelerating Rotarod test and body weights at baseline and 30 d after 5 × 0.1 ng/g DTx. Mice were 13–16 weeks old at the onset of DTx. Sample size *n* = 4–6 per group.

**Figure 5 ijms-22-09496-f005:**
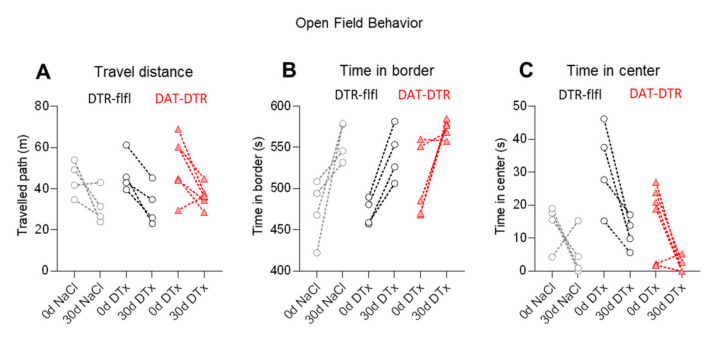
Paired analysis of test-retest behavior in the open field test. (**A**–**C**) Paired analysis of test-retest OFT behavior of individual mice was used to assess learning. Data were compared with 2-way ANOVA and revealed a significant effect of time (i.e., test versus retest) but not of the between-subject factor group. Sample sizes *n* = 4–6 per group.

**Figure 6 ijms-22-09496-f006:**
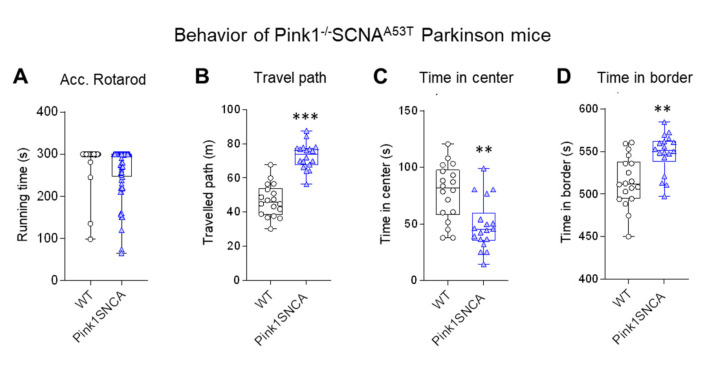
Behavior in genetic Parkinson Pink1^−/−^SNCA^A53T^ double mutant mice. (**A**) Running time on an accelerating Rotarod. (**B**–**D**) Travel paths and times spent in the center and border compartment in the open field test (OFT). Pink1^−/−^SNCA^A53T^ were hyperactively running along the walls. Mice were 12–15 months of age. Each scatter is a mouse; *n* = 18 per group. Data were compared with *t*-tests; asterisks indicate significant differences between genotypes.

**Table 1 ijms-22-09496-t001:** Examples of DTx-based mouse models targeting neurons or glia of the central nervous system.

Mice Expressing DTR	DTx Dosing and Schedules	DTx Effects	Reference
Dopaminergic neurons: DAT-DTR (Slc6a3^DTR/+^)	50 ng/g DTx s.c. at the age of 3–5 days DTx: List Biological Laboratories	Loss of TH+ neurons. Impaired cognition with deficits in spatial learning, spatial memory and object recognition. Impaired motor coordination on balance beam and rotarod	[44]
Doublecortin-positive neurons: DCX-DTR generated by insertion of DTR via homologous recombination	10 ng/g DTx per day for 10 d i.p.	Deficits of spatial learning and reversal learning	[45]
Oligodendroglia: MOG-Cre; Rosa26 loxP-STOP-loxP DTR (= R26-LSL-DTR)	100 ng DTx per mouse (≈4 ng/g) 1 × daily for 3 d, and 3 × daily for 7 d	DTx evoked myelin loss and white matter CNS pathology, tremor, hind limb paralysis and BW loss after 30 d.	[33]
Oligodendroglia: MOG-Cre; R26-LSL-DTR	400 ng/mouse (≈16 ng/g) DTx in PBS once daily for seven days i.p. DTx: Merck	Depletion of OGC-induced axonal injury, but did not cause neuronal cell death	[46]
ETS domain factor-positive 5HT neurons in adult mice: Pet1-Cre; R26-LSL-DTR	5, 50 and 250 ng/g DTx i.p. in saline; 1, 3, 5 times per week for 1, 3 or 6 weeks Cumulative dose of DTx ranged from 2 to 35 µg. DTx: Sigma DO564	DTx-evoked reduction of 5-HT neurons (ca. 80%). Nonspecific effectsand increase in mortality at high cumulative dose. Drop of body temperature at 1 week	[41]
5HT neurons: Pet1-Cre; R26-LSL-DTR	20 ng/g DTx i.p. 1 × /d for two days DTx: Sigma	Enhanced dendritic length of newborn hippocampal neurons	[47]
Microglia: Tamoxifen-inducible CX3CR-Cre^ERT^; R26-LSL-DTR	2 × 2 mg tamoxifen s.c. at the age of 12–14 days to induce DTR expression in microglia 500 ng DTx i.p. per mouse (≈20 ng/g) once daily for 3 consecutive days at the age of 8 weeks DTx: Merck Millipore	Ablation of 80% of microglia in the brain and the spinal cord 3 days after DTx injection	[48]
Microglia: Tamoxifen-inducible CX3CR-Cre^ERT^; R26-LSL-DTR	1 µg/mouse (≈40 ng/g) i.p. for 3 consecutive days DTx: Sigma	Deficits in multiple learning tasks and reduction in motor learning-dependent synapse formation	[42]
Agouti-related protein-positive neurons: AgRP-Cre; R26-LSL-DTR	DTx i.p. in neonates or adult mice	Ablation of NPY/AgRP neurons in adult mice resulted in rapid starvation.	[39]
Agouti-related protein-positive neurons: AgRP-Cre; R26-LSL-DTR	50 ng/g 2 × i.m. 2d apart in 6-week-old mice DTx: List Biological Laboratories	Ablation of AgRP neurons led to increase of cFOS in subsets of neurons and gliosis	[40]
Itch sensing neurons: Advillin-Cre; R26-LSL-DTR and subsequently Somatostatin Sst^Cre^ mice X Advillin-DTR to produce heterozygote Sst-Cre: Avil-Cre-DTR/+ mice	40 ng/g of DTx, 2 injections, 3 days apart	Reduced scratching behavior evoked by interleukin-31 or agonist at the 5HT1F receptor	[49]
Glucose transporter Glut4-Cre; R26-LSL-DTR Pro-opiomelanocortin Pomc-Cre; R26-LSL-DTR	Stereotaxic hypothalamic injection of 4 ng DTx/mouse (≈0.16 ng/g)	Anorexia in Glut4-DTR mice, hyperphagia in Pomc-DTR mice	[50]
Single minded-1-positive hypothalamic neurons: Sim1-Cre; R26-LSL-DTR	Intra-cerebro-ventricular (ICV) 2.5 ng DTx/mouse in 2 µL artificial CSF (≈0.1 ng/g)	Ablation of Sim1-neurons resulted in obesity	[51]
Tyrosine hydroxylase-positive neurons: TH-Cre; R26-LSL-DTR	Pegylated DTx (Calbiochem) 0.02 pmol/g once daily for 8 consecutive days	PEGy-DTx led to regional ablation of sympathetic neurons. Pegylation prevented crossing the blood-brain barrier	[43]

Cre/loxP models were generated by crossing the respective Cre-mouse with a mouse carrying DTR headed with a loxP-STOP-loxP site (LSL-site). The construct was inserted into the Rosa26 locus (R26-LSL-DTR). Cre-recombinase excised the STOP codon, leading to DTR expression in Cre-positive (Cre+) cells.

## Data Availability

The data generated for this manuscript are presented within the manuscript or in Appendix A. Raw data shall be made available on reasonable request.

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
