# Peer review of "Failure of Diphtheria Toxin Model to Induce Parkinson-Like Behavior in Mice"

_ijms, 2021, doi:10.3390/ijms22179496_

Round 1

Reviewer 1 Report

In this manuscript, the authors developed a novel conditional in vivo model for the study of Parkinson’s disease expressing transgenic diphtheria toxin receptor but it failed to recapitulate the disorder behaviour. I think that this is a very interesting study and I have only some concerns:

  • How many animals did you use? Please put the number in the leg and of each figure
  • I suggest the authors to better discuss the potentiality of their model in the Conclusion section
  • Please correct some grammar errors and mistyping

Author Response

Reviewer #1

In this manuscript, the authors developed a novel conditional in vivo model for the study of Parkinson’s disease expressing transgenic diphtheria toxin receptor but it failed to recapitulate the disorder behaviour. I think that this is a very interesting study and I have only some concerns:

How many animals did you use? Please put the number in the leg and of each figure

I suggest the authors to better discuss the potentiality of their model in the Conclusion section

Please correct some grammar errors and mistyping

Response

Thank you for evaluation of our manuscript and your suggestions.

We have added sample sizes in the legends as requested. Changes in the manuscript are highlighted by WORD's track changes. The data are presented as scatter plots to reveal results of individual animals, or immunofluorescence images of individual mice.

We have expanded the Conclusion as suggested.

Few typos were found and corrected. 

Reviewer 2 Report

The paper by Valek et al. analyzed potential role of diphteria toxins to induce Parkinson-like behavior in mice. The presented topic is really important, there is urgent need of new models in Parkinson disease. Despite the study is negative, the methodology looks correct. They use proper language, the paper is very clear and easy to read. The paper definitely fits to the scopus of the IJMS. I have only minor suggestion to add in introduction section some more data about new trends in pathophysiology of synucleinopathies and potential usage as a model (please use Wernick et al. Clin Autonom Res. 2021 Feb;31(1):117-125.

Author Response

Reviewer #2

The paper by Valek et al. analyzed potential role of diphteria toxins to induce Parkinson-like behavior in mice. The presented topic is really important, there is urgent need of new models in Parkinson disease. Despite the study is negative, the methodology looks correct. They use proper language, the paper is very clear and easy to read. The paper definitely fits to the scopus of the IJMS. I have only minor suggestion to add in introduction section some more data about new trends in pathophysiology of synucleinopathies and potential usage as a model (please use Wernick et al. Clin Autonom Res. 2021 Feb;31(1):117-125.

Response

Thank you for evaluation of our manuscript and positive feedback.

We have added a short paragraph about synucleopathies and multiple system atrophy including the suggested reference.

Reviewer 3 Report

The authors present the results of the study of diphteria toxin specific damage of dopaminergic neurons as a possible parkinson disease model; they found tirosine hidroxylase decreased mRNA expression in this neurons although there was no in vivo correlate with parkinson disease (PD). They argue that the toxin doses were not enough to generate the PD model, but higher doses had high letality and could not be used.

Pharmaceutical quality controls of parenteral products are strict. Preparations for injections must have several parameters, and chemical supplies are not frequently prepared for this matter. Lethality should appear from the injection, even though authors searched for bacterial contamination only. This factor does not permit a clear conclusion of the controled damage of dopaminergic neurons, although the model should be refined to be useful as a PD model in mice. Authors should have argue about the novelty of the model compared with the other in vivo models that involve DA neurons damage; they specified the lack of synuclein aggregates in some of the models but they didn´t delve in the relevance of the features for their proposal. They described appropriately the results on the TH findings related to the DTx doses.

English editing is needed in some details, for instance some orthographic mistakes (does – doses; one months); references order for table 1 can be improved.

Author Response

Reviewer #3

The authors present the results of the study of diphteria toxin specific damage of dopaminergic neurons as a possible parkinson disease model; they found tirosine hidroxylase decreased mRNA expression in this neurons although there was no in vivo correlate with parkinson disease (PD). They argue that the toxin doses were not enough to generate the PD model, but higher doses had high letality and could not be used.

Pharmaceutical quality controls of parenteral products are strict. Preparations for injections must have several parameters, and chemical supplies are not frequently prepared for this matter. Lethality should appear from the injection, even though authors searched for bacterial contamination only. This factor does not permit a clear conclusion of the controled damage of dopaminergic neurons, although the model should be refined to be useful as a PD model in mice. Authors should have argue about the novelty of the model compared with the other in vivo models that involve DA neurons damage; they specified the lack of synuclein aggregates in some of the models but they didn´t delve in the relevance of the features for their proposal. They described appropriately the results on the TH findings related to the DTx doses.

Response

Thank you for evaluation of our manuscript and your suggestions.

The lyophilized powder of DTx (Sigma) was reconstituted in sterile water to obtain the stock solution, and was then further diluted in 0.9% sterile saline for injections in mice. There were no chemical involved and vehicle was handled identically. We have added this information in the Methods. It is also described in the legend of Suppl. Table S2.

The theoretical advantages of the flexible Cre/loxP-DTR model in comparison with genetic models and the toxic models is outlined in the Introduction (line 58-63 and 84 pp). The aim was to study putative advantages, particularly the non-invasive, non-hazardous slowly progressive and dose-dependent decline of DA neurons, which was partly achieved and in parallel the development of PD-like measurable in vivo phenotypes. The latter was not achieved. Hence, we have to conclude that the model did not fulfil the expectations, was far more difficult than expected and was overall a disappointment.

English editing is needed in some details, for instance some orthographic mistakes (does – doses; one months); references order for table 1 can be improved.

We have corrected the typing errors.

The references are formatted according to Journal style, i.e. the reference gets a number on first occurrence in the text including the table. The table is sorted and grouped according to the genetic models and targeted cells of the respective studies.